# Analysis and Prediction of Electrospun Nanofiber Diameter Based on Artificial Neural Network

**DOI:** 10.3390/polym15132813

**Published:** 2023-06-25

**Authors:** Ming Ma, Huchen Zhou, Suhan Gao, Nan Li, Wenjuan Guo, Zhao Dai

**Affiliations:** 1School of Life Sciences, Tiangong University, Tianjin 300387, China; 2State Key Laboratory of Separation Membranes and Membrane Processes, Tiangong University, Tianjin 300387, China; 3School of Chemical Engineering and Technology, Tiangong University, Tianjin 300387, China; 4School of Chemistry, Tiangong University, Tianjin 300387, China; 5School of Pharmaceutical Sciences, Tiangong University, Tianjin 300387, China

**Keywords:** electrostatic spinning, artificial neural network, diameter, prediction, nanofibers

## Abstract

Electrospinning technology enables the fabrication of electrospun nanofibers with exceptional properties, which are highly influenced by their diameter. This work focuses on the electrospinning of polyacrylonitrile (PAN) to obtain PAN nanofibers under different processing conditions. The morphology and size of the resulting PAN nanofibers were characterized using scanning electron microscopy (SEM), and the corresponding diameter data were measured using Nano Measure 1.2 software. The processing conditions and corresponding nanofiber diameter data were then inputted into an artificial neural network (ANN) to establish the relationship between the electrospinning process parameters (polymer concentration, applied voltage, collecting distance, and solution flow rate), and the diameter of PAN nanofibers. The results indicate that the polymer concentration has the greatest influence on the diameter of PAN nanofibers. The developed neural network prediction model provides guidance for the preparation of PAN nanofibers with specific dimensions.

## 1. Introduction

In recent years, nanofibers, as a distinctive nanomaterial, have gained considerable attention [1,2]. Compared to conventional materials, nanofibers commonly exhibit high surface area to volume ratio, high porosity, and exceptional mechanical strength, thus making them a promising material in various fields, such as catalysis [3], sensing [4], and adsorption [5]. As an important nanomaterial, polyacrylonitrile (PAN) nanofibers have garnered significant attention due to their high surface area to volume ratio, high porosity, good biocompatibility and biodegradability, and exceptional mechanical strength [6]. They are widely used in various fields such as tissue engineering, drug delivery, water treatment, and air purification [1,2,7]. PAN nanofibers can be prepared by various methods including electrospinning [7,8], centrifugal spinning [8], and stretching spinning [8]. Among these techniques, electrospinning is widely employed for PAN nanofiber preparation, owing to its simplicity, efficiency, and good controllability [7,9].

Although electrospinning is a widely used method for fabricating PAN nanofibers, achieving consistent fiber diameter and desired properties can be challenging [10,11]. One of the main difficulties is controlling the process parameters, including solution properties, electric field, and collecting distance, to obtain optimal fiber morphology and properties. The viscosity and conductivity of the PAN solution may also be carefully controlled to avoid issues such as bead formation, fiber breakage, and inconsistent fiber diameter. Other factors that can impact the electrospinning process comprise humidity, temperature, and air flow, which can affect fiber alignment and orientation [10,11,12,13,14,15]. Overall, the successful electrospinning of PAN nanofibers requires thorough optimization of multiple process parameters and attention to detail to ensure reproducibility and consistent fiber properties. To overcome this challenge, researchers have been investigating the effect of various process parameters on the diameter and properties of PAN nanofibers. By comprehending the impact of these factors on the properties of PAN nanofibers, researchers hope to improve the production process and prepare nanofibers with the desired characteristics. Zhang et al. selectively adjusted the solution properties by adding iron acetylacetonate to the electrospinning solution in order to control the diameter of PAN fibers [16].

A promising approach for optimizing the electrospinning process to produce consistent PAN nanofibers is the use of artificial neural networks (ANNs), a type of machine learning model [17]. ANNs consist of multiple layers of interconnected nodes that can process and learn from data and are designed to mimic the behavior of the human brain, where neurons communicate through electrical and chemical signals. In ANNs, each node receives input signals and generates an output signal based on a set of weights and biases, which are updated during the training process to minimize the difference between predicted and actual output. To use ANNs for electrospinning optimization, researchers first need to collect experimental data on the relationship between process parameters and fiber properties, such as fiber diameter, porosity, and mechanical properties. This data can be used to train the ANN to recognize patterns and make predictions under different process conditions, such as varying solution concentration, electric field strength, and collecting distance. Once trained on this data, the ANN can predict the properties of PAN nanofibers produced under new process conditions by taking in the process parameters as input and generating a prediction of the resulting fiber properties. By analyzing the output of the ANN, researchers can determine which process parameters have the greatest impact on fiber properties and identify the optimal values for these parameters [17,18,19,20]. Faridi-Majidi et al. utilized the ANN method to predict the impact of different needleless electrospinning parameters on the diameter of PAN nanofibers by considering parameters such as polymer solution concentration, applied voltage, and nozzle-to-collector distance [17]. Gari et al. trained an ANN model using the backpropagation algorithm, with average prediction errors of fiber diameter found to be 0.05% and 2.6% for training and testing data, respectively [18]. Compared to traditional optimization methods such as trial-and-error experimentation and statistical modeling, ANNs can handle large amounts of data and identify nonlinear relationships between process parameters and fiber properties that may not be captured by other methods. Additionally, ANNs can continuously learn and improve over time, making them well-suited for dynamic and complex systems like electrospinning [21,22,23,24]. Overall, the use of ANNs in electrospinning optimization has the potential to accelerate the development of new PAN nanofiber materials with tailored properties for various applications.

Based on the above, this study utilized electrospinning technology to fabricate PAN nanofibers with different diameters. The morphology and size of the produced PAN nanofibers were characterized using scanning electron microscopy (SEM), and the corresponding diameter data were measured. An ANN model was trained using diameter and process parameter data to develop an accurate PAN nanofiber diameter prediction model. Additionally, we aimed to address the limitations of previous studies by providing a more comprehensive analysis of the impact of electrospinning process parameters on nanofiber diameter. In contrast to the existing references, our study not only investigates the individual effects of polymer concentration, applied voltage, collecting distance, and solution flow rate on PAN nanofiber diameter but also explores their combined and non-linear influences. This approach enables us to gain a deeper understanding of the intricate relationship between process parameters and nanofiber diameter. Furthermore, our research goes beyond the prediction model and provides guidance for the preparation of PAN nanofibers with specific dimensions based on the established neural network prediction model. By considering the unique advantages of our study, we contribute to the advancement of the field of nanofiber synthesis and characterization.

## 2. Experimental Section

### 2.1. Materials and Reagents

Polyacrylonitrile (PAN) was furnished by Suzhou Hui Huang Plasticizing Co. (Suzhou, China). Tianjin Kermel Chemical Reagent Co. Ltd. (Tianjin, China) offered N, N-dimethylformamide (DMF).

### 2.2. Fabrication of PAN Electrospinning Solution

PAN (Mw = 85,000 g/mol) were prepared at concentrations of 6 wt%, 8 wt%, 10 wt%, 12 wt%, 14 wt%, and 16 wt%. To prepare the solutions, 9 mL of DMF was accurately transferred to a transparent glass bottle using a 1000 μL pipette. For each concentration, 0.5443 g, 0.7500 g, 0.9475 g, 1.1598 g, 1.3845 g, and 1.6200 g of PAN solid powder were weighed and added to the bottle with an appropriate-sized magnetic stir bar. The mixture was stirred until the PAN solid powder was completely dispersed. The bottle was then transferred to a magnetic hotplate stirrer with a constant temperature water bath at 80 °C and stirred at 1000 rpm for 10 h. The prepared spinning solution was used as soon as possible.

### 2.3. Fabrication of PAN Nanofiber

The PAN electrospinning solution was prepared and loaded into a 10 mL syringe. The syringe was connected to the spinneret needle (22-gauge) via a Luer fitting and a transparent plastic tube. The syringe was fixed onto an injection pump, and the spinneret needle was connected to the positive electrode of a high-voltage power supply (the negative electrode was fixed to the collector). The electrospinning voltage, distance between the spinneret and the collector, and flow rate of the spinning solution were optimized. Under the influence of the electrostatic field, the spinning solution was sprayed onto the collector, resulting in the formation of PAN nanofibers. The electrospinning process was conducted at 25 ± 5 °C and a relative humidity of 30%. The obtained PAN nanofibers were vacuum dried at 50 °C and stored for future use.

### 2.4. Characterization of PAN Nanofiber

Scanning electron microscope (SEM, Geminutesi SEM500, Oberkochen, Germany) was used to observe the surface morphology, structure, and size of the nanofibers. In order to characterize the morphology and dimensions of the PAN nanofibers produced by electrospinning, a subset of 100 nanofibers from the SEM images were selected for diameter measurement using the Nano Measure 1.2 software. The average diameter was calculated and used as a representative value for the sample. The measured diameter values were then analyzed in conjunction with the corresponding electrospinning parameters to gain insight into the relationship between process conditions and nanofiber morphology. This information is crucial for optimizing the electrospinning process and tailoring the properties of the resulting nanofibers for various applications.

### 2.5. Experimental Data Set

The 137 sets of PAN nanofiber data samples obtained from the electrospinning experiments were used to construct an artificial neural network. In this dataset, the concentration of the PAN spinning solution ranged from 6 wt% to 16 wt%, the spinning voltage ranged from 10 kV to 22 kV, the change range of the receiving distance was from 12 cm to 22 cm, and the injection speed of the spinning solution ranged from 0.1 mL/h to 0.5 mL/h. The PAN spinning process parameters and corresponding nanofiber diameters contained in each data group are shown in Appendix A in the Appendix A.

The PAN spinning parameters involved, including PAN spinning solution concentration, spinning voltage, receiving distance, injection rate, and the resulting PAN nanofiber diameter, have different units, which may affect the prediction results of the ANN model and lead to significant errors. Thus, data normalization preprocessing is necessary. Common normalization techniques comprise maximum and minimum value normalization, logarithmic function normalization, and standard normalization. In this study, we employed the logarithmic function normalization method to preprocess the data, enabling all data to have the same impact scale on the model.

In artificial neural networks, all input data is usually divided into three parts: training set, validation set, and test set. The training set was used to train the network model, the validation set was used to check the effectiveness of the network model, and the test set was used to test the final performance of the network model. The partitioning of the dataset plays an essential role in model establishment. When the dataset is improperly partitioned, the model may experience overfitting, where the training data fits the model well, but the test results are not satisfactory. Therefore, it is necessary to partition the dataset reasonably. As the test set does not affect the construction of the network model, after removing unsuitable data groups, the data was divided into three groups using the hold-out method. The 137 sample data sets were randomly divided into three groups with a test set ratio of 0, a training set ratio of 0.8, and a validation set ratio of 0.2.

### 2.6. Training of the Artificial Neural Network

The artificial neural network algorithm hierarchy mainly includes the input layer, hidden layer, and output layer. As shown in Figure 1, the spinning solution concentration, spinning voltage, receiving distance, and injection rate are set as the input layer, the weight allocation and integration of the relationship between spinning parameters are reflected in the hidden layer, and the nanofiber diameter is the output layer [22].

The number of hidden layers is generally more than one, and the specific number of layers is determined by the specific requirements of the problem and the number of nodes. When the number of hidden layers is too small, the performance of the trained network model is poor, or the network may not even be formed. However, when the number of hidden layers is too large, the learning time of the network will be too long, and the resulting network model results will also vary. In this chapter, we set the number of hidden layers to four, with each hidden layer containing 20 neural nodes. In the neural network, the Levenberg–Marquardt algorithm, which has a fast convergence speed and strong training ability, is called to train the data, and the function statement is “trainFcn = ‘trainlm’.” The input parameters are optimized using a genetic algorithm [25].

The training parameters are set after the neural network is created. The training parameters include the number of training times, the training target to be achieved, and the learning rate of the network. Generally, the trial and error method is used to determine the number of training times. Here, we set the initial training times to 500 cycles and modified them according to the training results of the model. The learning rate was set to 0.1. A too-small learning rate will prolong the training time of the network, while a too-large learning rate will affect the convergence results of the network. The training goal of the neural network is to minimize the difference between the simulated training prediction values and the actual values. The difference between the predicted nanofiber diameter and the actual diameter can be compared using the Pearson correlation coefficient R and the root mean square error [25,26,27,28,29,30].
(1)$$R=\frac{cov\left(X,Y\right)}{\sigma X\sigma Y}=\frac{E\left(X,Y\right)-E\left(X\right)E\left(Y\right)}{\sqrt{E\left(X^{2}\right)-E^{2}\left(X\right)}\sqrt{E\left(Y^{2}\right)-E^{2}\left(Y\right)}}$$
(2)$${ANN}_{RSME}=\sqrt{\frac{1}{n}\sum_{i=1}^{n}{\left(Y-X\right)}^{2}}$$
where *X* represents the actual diameter of the nanofiber, and *Y* represents the predicted diameter of the nanofiber.

When the model starts training, the weight allocation values on each neuron will be continuously adjusted until the measured data reaches the required range, and the training will stop running, indicating the end of the training.

## 3. Result and Discussion

### 3.1. Morphology Characterization of Electrospun PAN Fiber

A small quantity of vacuum-dried PAN nanofiber samples were immobilized on a sample stage, followed by surface coating with gold before visualizing their morphology and size in SEM. The obtained images are presented in Figure 2. A uniform and well-distributed PAN nanofiber with a diameter of 276.03 nm was prepared when the PAN concentration was 12 wt%, the spinning voltage was 14 kV, the collecting distance was 20 cm, and the feeding rate was 0.4 mL/h.

### 3.2. Analysis of the Artificial Neural Network Model

After adjusting the training parameters and setting the number of iterations to 1500, a fairly good artificial neural network (ANN) model was obtained. The R coefficient, i.e., the Pearson correlation coefficient, reflects the fitting degree of the ANN model to the relationship between the various parameters and the dependent variable, the nanofiber diameter. Through weight adjustment and model training, as shown in Figure 3, the ANN algorithm model showed a high degree of fit for the training set with a fit coefficient close to 1, a fit coefficient of 0.95287 for the validation set, and a fit coefficient of 0.98952 for the comprehensive data set. The fitting curve of the output value and the target value is excellent, indicating that the ANN model has excellent prediction capability.

The measured diameter of PAN nanofibers and the corresponding diameter simulated by the ANN model are illustrated in Figure 4. It is evident from the figure that the measured fiber diameter is in excellent agreement with the simulated fiber diameter obtained by the model. Figure 5 shows the error between the measured diameter of PAN nanofibers and the simulated diameter corresponding to the established artificial neural network model. It can be observed from the figure that the error between the measured diameter and simulated diameter of all data samples is less than 10%, and the calculated average error is 2%.

To verify the predictive accuracy of the constructed neural network model, PAN nanofiber electrospinning data that were not used in the model training were imported into the prediction module of the neural network to obtain the simulated diameter of PAN nanofibers, and the error was calculated. The results are shown in Table 1. It can be concluded that the ANN model exhibits a considerable predictive accuracy for the diameter of PAN electrospun nanofibers.

### 3.3. Interactions among Electrospinning Process Parameters

The influence of electrospinning process parameters, including polymer concentration (spinning solution concentration), applied electrostatic voltage (spinning voltage), distance between the needle tip and collector (collection distance), and the flow rate of the polymer solution (feeding rate), on the diameter of electrospun nanofibers is complex and non-linear. Therefore, it is necessary to investigate how different factors affect the diameter of nanofibers. In this study, we focus on the combined effects of the electrospinning process parameters on the diameter of PAN nanofibers. Electrospinning process parameters are varied within a certain range in an arithmetic sequence and imported into the established artificial neural network. The neural network model predicts the diameter of PAN nanofibers corresponding to the input parameters, and the data is plotted to obtain a three-dimensional visualization of the results.

Figure 6 shows the three-dimensional visualization of the prediction results of PAN nanofiber diameter by the ANN model under the joint effect of spinning solution concentration and spinning voltage. The receiving distance was fixed at 20 cm, and the injection speed was 0.3 mL/h. The spinning solution concentration varied evenly in the range of [6 wt%, 16 wt%], with a step of 2 wt%; the spinning voltage varied evenly in the range of [10 kV, 22 kV], with a step of 2 kV. It can be observed from the figure that the spinning solution concentration has a greater influence on the diameter of PAN nanofibers than the spinning voltage. As the spinning solution concentration increases, the diameter of PAN nanofibers also increases. When the spinning solution concentration is low, the diameter of PAN nanofibers first increases and then slightly decreases and tends to be flat with the increase in spinning voltage. On the contrary, when the spinning solution concentration is high, the diameter of PAN nanofibers first decreases and then increases with the increase in spinning voltage. When both factors change together, the diameter of PAN nanofibers gradually increases with their increase.

Most studies have shown that increasing the voltage typically leads to a decrease in fiber diameter, and the influence of voltage is relatively smaller compared to concentration and infusion rate. However, there are also some exceptional cases where studies have shown that increasing the voltage can result in an increase in fiber diameter. For example, literature on cellulose [31], gelation [32], and zein [33] has reported such observations. Additionally, a study on polyurethane [34] has demonstrated that increasing the voltage can also lead to an increase in jet diameter, thus affecting the final fiber diameter. Therefore, we believe that the impact of increasing voltage on fiber diameter is a complex phenomenon influenced by multiple factors. Firstly, increasing voltage generally leads to a decrease in jet diameter, thereby reducing fiber diameter. Secondly, as demonstrated by studies on cellulose, lower voltage contributes to longer flight time of the jet, facilitating further splitting into smaller fibers during flight, contrary to the commonly held belief that reducing voltage increases fiber diameter. Our research investigates the comprehensive effects of electrospinning parameters, including voltage. Previous studies often fix other parameters and explore variations in a single parameter.

This observed behavior of the PAN nanofiber diameter under the joint influence of spinning solution concentration and spinning voltage can be explained by considering the interplay between these parameters. It is well-known in the standard electrospinning literature that increasing voltage and decreasing distance between the needle tip and collector result in decreased fiber diameters. Similarly, increasing polymer concentration generally leads to increased fiber diameters. In our study, we observed a similar trend when varying these parameters individually. However, when considering their combined effects, the situation becomes more complex. The interaction between spinning solution concentration and spinning voltage can lead to a non-linear response in the PAN nanofiber diameter. The increase in spinning solution concentration can contribute to the enlargement of fiber diameters, while the increase in spinning voltage alone tends to reduce the diameter. When both parameters increase simultaneously, the dominant effect of spinning solution concentration outweighs the influence of spinning voltage, resulting in an overall increase in the fiber diameter.

Figure 7 is a 3D plot of the prediction results of the PAN nanofiber diameter by the ANN model under the combined effects of spinning solution concentration and collector distance. Here, the spinning voltage is fixed at 16 kV, the flow rate is 0.3 mL/h, and the spinning solution concentration varies uniformly in the range of [6 wt%, 16 wt%] with a step size of 2 wt%, while the collector distance varies uniformly in the range of [12 cm, 22 cm] with a step size of 2 cm. From the figure, we can observe that the spinning solution concentration has a greater influence on the PAN nanofiber diameter compared to the collector distance. As the spinning solution concentration increases, the diameter of PAN nanofibers also increases overall. The impact of collector distance on fiber diameter is weaker. When the spinning solution concentration is 16 wt%, the diameter of PAN nanofibers increases gradually with increasing collector distance. When the collector distance is 18 cm, the diameter of PAN nanofibers first increases, then becomes flat, and then continues to increase with increasing spinning solution concentration. On the other hand, when the collector distance is shorter, the diameter of PAN nanofibers increases with increasing spinning solution concentration.

Figure 8 shows the 3D prediction results of the ANN model for the diameter of PAN nanofibers under the joint action of spinning solution concentration and injection rate. The spinning voltage is fixed at 16 kV, the receiving distance is fixed at 16 cm, the spinning solution concentration varies evenly in the range of [6 wt%, 16 wt%] with a step size of 2 wt%, and the injection rate varies evenly in the range of [0.1 mL/h, 0.5 mL/h] with a step size of 0.1 mL/h. From the graph, it can be observed that the effect of spinning solution concentration on the diameter of PAN nanofibers is relatively higher than that of the injection rate. When both concentration and injection rate increase, the diameter of PAN nanofibers also increases. When the injection rate is low, the diameter of PAN nanofibers shows a linear increase with the increase in spinning solution concentration. However, when the injection rate is 0.5 mL/h, the diameter of PAN nanofibers shows an initial increase, followed by a plateau, and then an increase with the increase in spinning solution concentration. When the concentration is low, the change in the diameter of PAN nanofibers is not significant with the variation of injection rate. Therefore, the effect of spinning solution concentration on the PAN nanofiber diameter is relatively higher than that of the injection rate. When both concentration and injection rate increase, the diameter of PAN nanofibers also increases. However, the specific response of the diameter to these parameters depends on their individual values.

Figure 9 illustrates the combined effects of spinning voltage and collector distance on the diameter of PAN nanofibers predicted by an artificial neural network model. The spinning solution concentration was fixed at 12 wt%, and the flow rate was fixed at 0.3 mL/h. The spinning voltage varied in equal increments from 10 kV to 22 kV, with a step size of 2 kV, and the collector distance varied in equal increments from 12 cm to 22 cm, with a step size of 2 cm. The figure shows that the joint effect of spinning voltage and collector distance on the diameter of PAN nanofibers is significant. When both factors increase, the diameter of PAN nanofibers first increases and then tends to level off. At a low spinning voltage, the diameter of PAN nanofibers gradually increases with increasing collector distance, while at a high spinning voltage, the opposite trend is observed. When the collector distance is small, the diameter of PAN nanofibers initially increases and then decreases with increasing spinning voltage.

Figure 10 illustrates the effect of the ANN model on the diameter of PAN nanofibers under the combined influence of spinning voltage and feed rate. Here, the spinning solution concentration was fixed at 12 wt%, the collection distance was fixed at 16 cm, and the spinning voltage was varied uniformly in the range of [10 kV, 22 kV] with a step size of 2 kV, while the feed rate was varied uniformly in the range of [0.1 mL/h, 0.5 mL/h] with a step size of 0.1 mL/h. From the graph, it can be observed that the diameter of PAN nanofibers gradually decreases as both the spinning voltage and collection distance increase. When the spinning voltage is low, the diameter of PAN nanofibers gradually increases with the increase in feed rate, while the effect of feed rate on the diameter of PAN nanofibers is not significant when the spinning voltage is high. When the feed rate is large, the diameter of PAN nanofibers increases first and then tends to plateau, with little change as the spinning voltage increases. However, when the spinning voltage is low, the diameter of PAN nanofibers gradually decreases with the increase in feed rate. Similarly, when the feed rate is low, the diameter of PAN nanofibers shows a decreasing trend with the increase in spinning voltage.

Figure 11 shows the effect of artificial neural network model on the diameter of PAN nanofibers under the combined action of collection distance and injection speed. The spinning solution concentration was fixed at 12 wt%, and the collection distance varied evenly in the range of [12 cm, 22 cm] with a step size of 2 cm, while the injection speed varied evenly in the range of [0.1 mL/h, 0.5 mL/h] with a step size of 0.1 mL/h. It can be seen from the figure that when the collection distance and injection speed increase together, the diameter of PAN nanofibers shows a gradually decreasing trend. When the collection distance is small, the diameter of PAN nanofibers gradually decreases with the increase in injection speed. When the collection distance is large, the effect of injection speed on the diameter of PAN nanofibers is consistent with the former. When the injection speed is high, the diameter of PAN nanofibers shows a trend of first increasing and then leveling off as the collection distance increases. When the injection speed is low, the diameter of PAN nanofibers first decreases and then increases gradually with the increase in collection distance.

Furthermore, we observed a significant combined effect of spinning voltage and collecting distance on fiber diameter. At lower spinning voltage, the fiber diameter increased with increasing collecting distance, while at higher spinning voltage, the fiber diameter decreased with increasing collecting distance. Additionally, we found that there was an interaction between spinning voltage and infusion rate in their effects on fiber diameter. At lower spinning voltage, the effect of infusion rate on fiber diameter was more pronounced, while at higher spinning voltage, the effect of infusion rate on fiber diameter was diminished. However, the interaction between collecting distance and infusion rate had a relatively minor effect on fiber diameter, particularly at higher infusion rates. Understanding these interaction effects can aid in optimizing electrospinning parameters to achieve the desired diameter of nanofibers.

In addition, the interactions among the four parameters, namely spinning solution concentration, spinning voltage, collecting distance, and infusion rate, complicate the impact on the diameter of PAN nanofibers. Firstly, spinning solution concentration and spinning voltage are two primary factors influencing fiber diameter. Higher spinning solution concentration typically results in larger fiber diameter, while higher spinning voltage leads to smaller fiber diameter, consistent with findings in standard electrospinning literature. Similar trends were observed when these two parameters were varied individually. However, when considering their combined effects, the situation becomes more complex. The interaction between spinning solution concentration and spinning voltage may result in a nonlinear response in PAN nanofiber diameter. Increasing spinning solution concentration leads to an increase in fiber diameter, while increasing spinning voltage causes a decrease in fiber diameter. However, when both parameters are increased simultaneously, the dominant effect of spinning solution concentration outweighs the influence of spinning voltage, resulting in an overall increase in fiber diameter. The impact of collecting distance and infusion rate on fiber diameter is relatively minor. Although they do have some effect on fiber diameter in certain cases, their influence is relatively subtle compared to spinning solution concentration and spinning voltage. The interactions among these parameters lead to a nonlinear response in fiber diameter, making it more complex. Through in-depth study and understanding of these interactions, we can better optimize the process parameters of electrospinning to obtain nanofibers with the desired diameter and performance.

## 4. Conclusions

In summary, we developed an accurate model for predicting the diameter of PAN nanofibers using ANN trained on diameter and process parameter data. The model achieved a high level of accuracy, with a fitting coefficient of 0.98952. Additionally, the study investigated the impact of process parameters, including spinning solution concentration, spinning voltage, receiving distance, and infusion rate on the diameter of the nanofibers. The results revealed that spinning solution concentration and spinning voltage had a significant effect on fiber diameter, while receiving distance and infusion rate had a relatively minor effect. This research provides valuable insights into the factors that affect PAN nanofiber diameter and could inform the optimization of spinning process parameters for the production of nanofibers with specific properties.

## Figures and Tables

**Figure 1 polymers-15-02813-f001:**
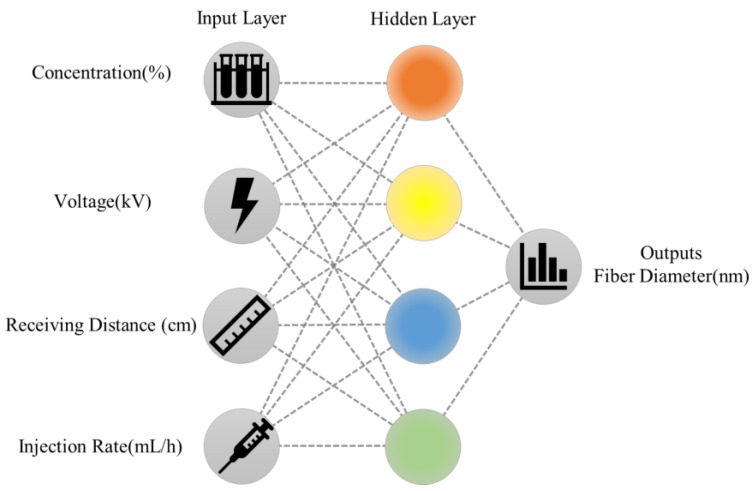
Schematic illustration of the simulating fiber diameter using artificial neural networks.

**Figure 2 polymers-15-02813-f002:**
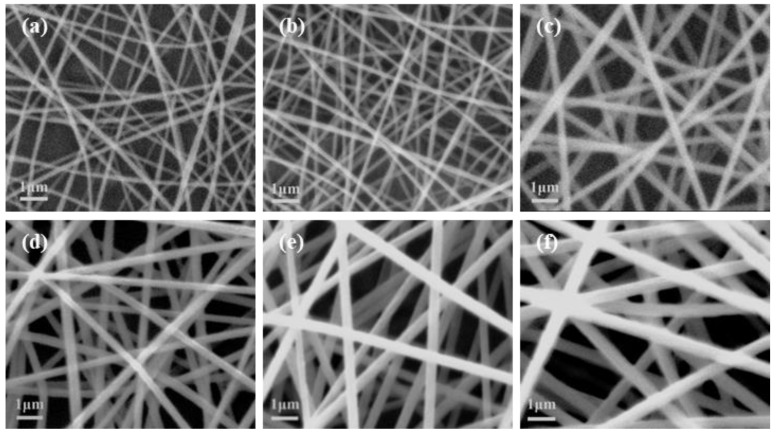
SEM images of PAN nanofiber with different concentrations, (**a**) 6 wt%, (**b**) 8 wt%, (**c**) 10 wt%, (**d**) 12 wt%, (**e**) 14 wt%, (**f**) 16 wt%.

**Figure 3 polymers-15-02813-f003:**
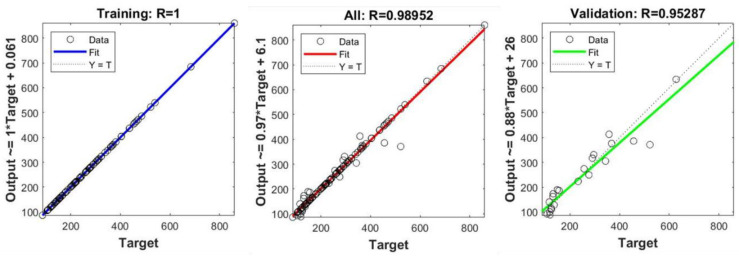
Fitting results of R coefficient.

**Figure 4 polymers-15-02813-f004:**
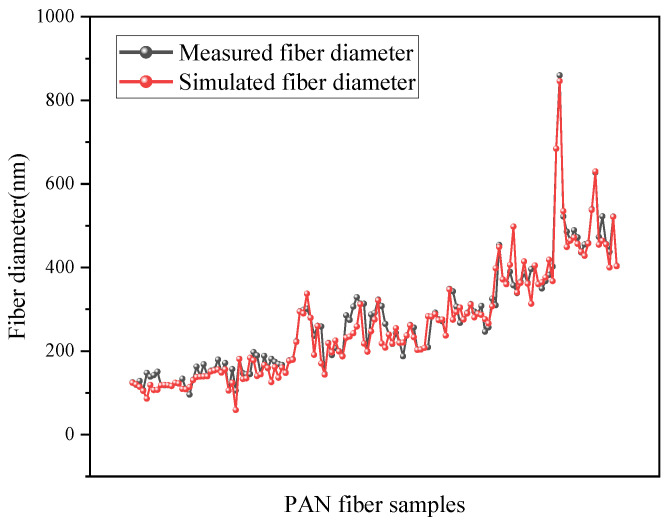
Measured fiber diameter and the simulated diameter by the neural network model.

**Figure 5 polymers-15-02813-f005:**
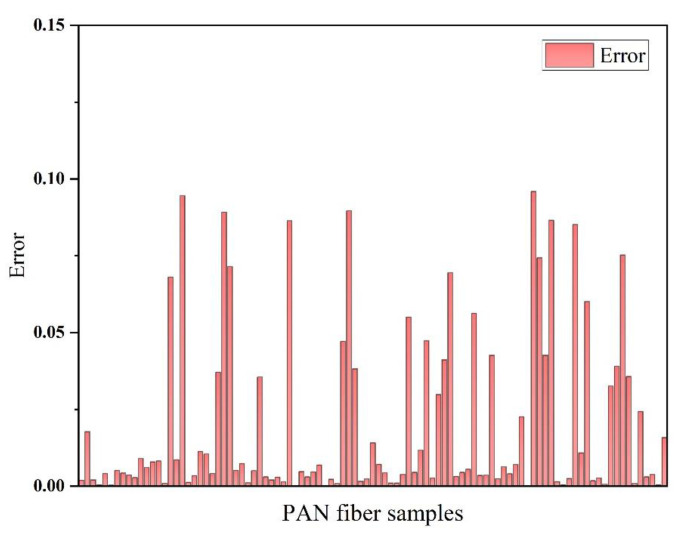
The error between the measured fiber diameter and the simulated diameter of the model.

**Figure 6 polymers-15-02813-f006:**
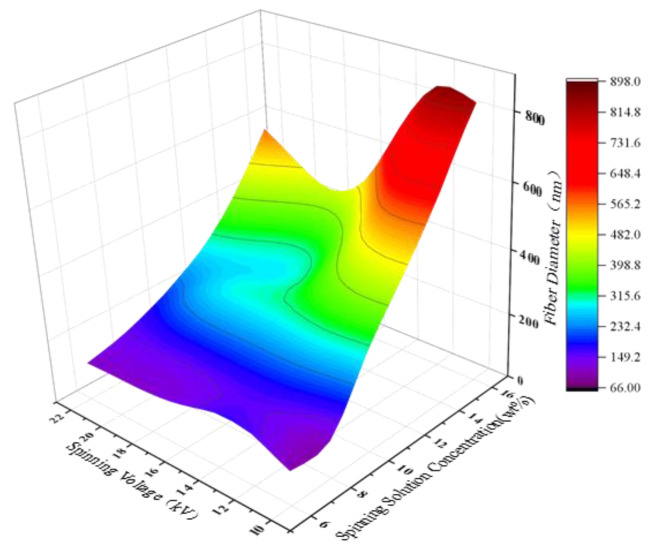
3D plot of the prediction results of PAN nanofiber diameter by ANN model under the joint action of spinning solution concentration and spinning voltage.

**Figure 7 polymers-15-02813-f007:**
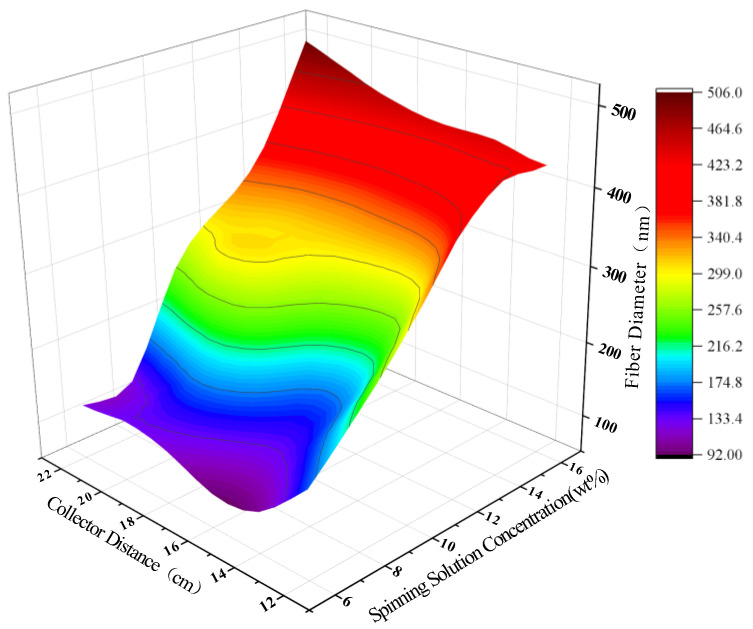
3D plot of the prediction results of PAN nanofiber diameter by ANN model under the joint action of spinning solution concentration and collector distance.

**Figure 8 polymers-15-02813-f008:**
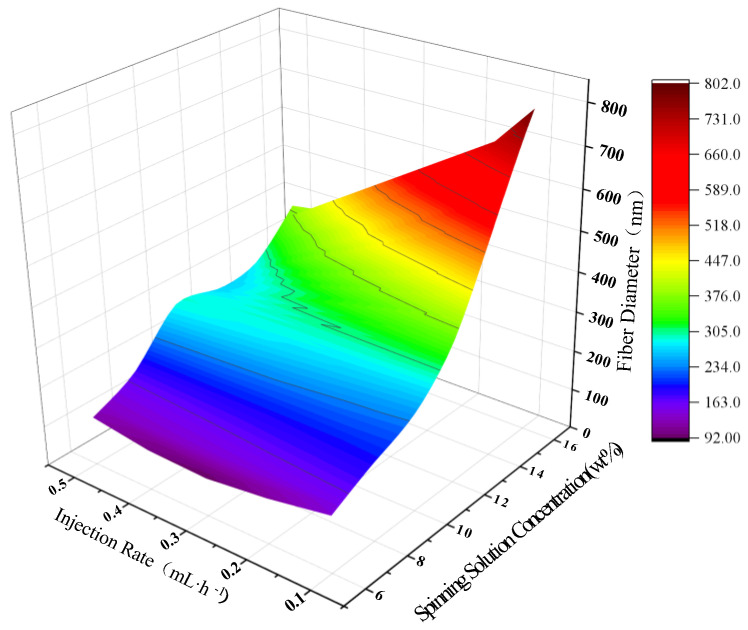
3D plot of the prediction results of PAN nanofiber diameter by ANN model under the joint action of spinning solution concentration and injection rate.

**Figure 9 polymers-15-02813-f009:**
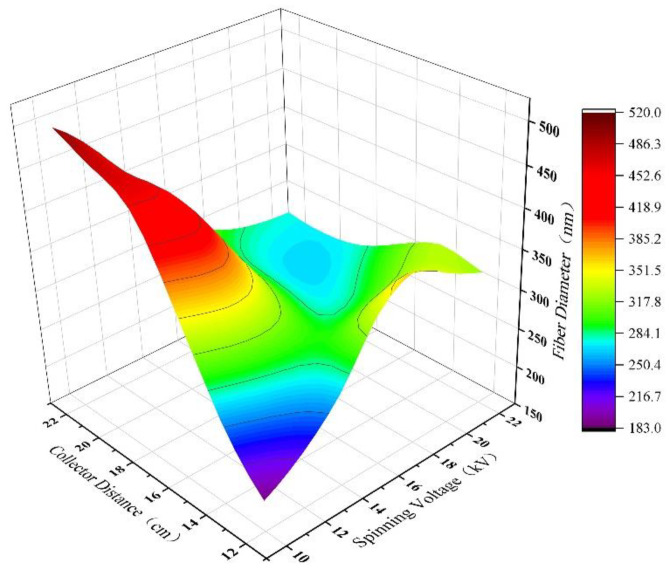
3D plot of the prediction results of PAN nanofiber diameter by ANN model under the joint action of spinning voltage and collector distance.

**Figure 10 polymers-15-02813-f010:**
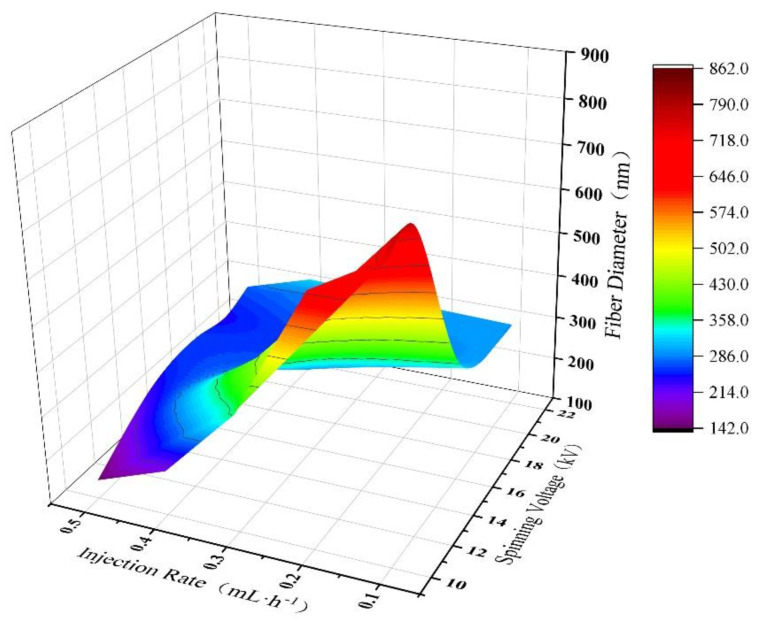
3D plot of the prediction results of PAN nanofiber diameter by ANN model under the joint action of spinning voltage and injection rate.

**Figure 11 polymers-15-02813-f011:**
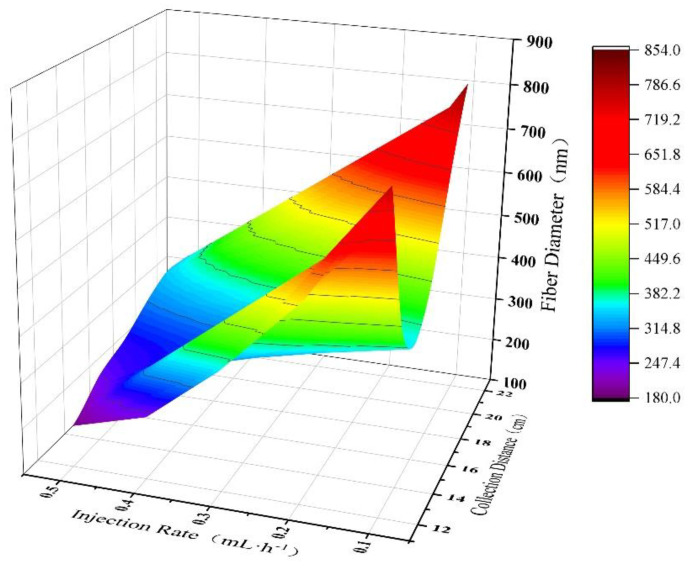
3D plot of the prediction results of PAN nanofiber diameter by ANN model under the joint action of collection distance and injection rate.

**Table 1 polymers-15-02813-t001:** Measured diameter of untrained PAN nanofibers and ANN predicted results.

Exp. No.	Concentration (wt%)	Voltage (kV)	Receiving Distance (cm)	Injection Speed (mL/h)	Measured Diameter (nm)	Simulated Diameter (nm)	Error
1	10	15	12	0.3	279.95	280.43	0.17%
2	10	15	12	0.4	295.02	299.22	1.42%
3	10	15	14	0.5	195.9	193.95	0.99%
4	10	15	16	0.3	200.28	202.26	0.99%
5	10	15	16	0.4	212.24	213.96	0.81%
6	10	15	18	0.2	173.99	173.41	0.33%
7	10	15	18	0.5	213.19	208.61	2.15%
8	10	15	20	0.2	235.15	237.61	1.05%
9	10	15	20	0.3	307.22	308.04	0.27%
10	10	15	20	0.5	165.93	169.51	2.16%

## Data Availability

The data presented in this study are available on request from the corresponding author.

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
