# Peer review of "Analysis and Prediction of Electrospun Nanofiber Diameter Based on Artificial Neural Network"

_polymers, 2023, doi:10.3390/polym15132813_

Round 1
Reviewer 1 Report
The manuscript by Ma et al. describes the analysis and prediction of e-spun PAN nanofiber diameters with a trained ANN. The manuscript is well-written and straight-forward, using systematically varied e-spinning parameter set of polymer conc., applied voltage, collector distance, and solution flow rate to produce the nanofiber meshes. The corresponding fiber diameter values were then fed into an ANN that was trained with the interaction matrix of the parameters, to evaluate the influence of each parameter setting on the fiber diameter.
Although the approach using an ANN to model & predict the corresponding fiber diameters, the weak point of the manuscript is that the Results & Discussion section mainly contains the description of the resulting 3D plots of the prediction results, but neither a thorough discussion nor a comparison with the results of other/the cited refs. Moreover, the authors miss out a discussion if the parameter interactions are reasonable, e.g. increasing voltage and increasing fiber diameter. Standard e-spinning literature teaches that increasing voltage and distance results in decreasing fiber diameters, whereas increasing polymer conc. and flow rate gives higher diameter values.
Minor points:
- Exp. section/2.3: How was the rel. humidity of 30% set or measured?
- Results & Discussion/3.3: Why do the ANN fitting results of „validation set“ and „comprehensive data set“ display deviations around 400-500 nm target values?
- Fig. 9, 10, 11: Scaling directions seem to be flipped, compared to the other figs.?
The overall quality of English language is good, and only needs some minor changes, mainly in the Introduction section and in the Abstract.
Reviewer 2 Report
I am in favor of publishing this manuscript containing interesting data and relevant analysis. I would suggest the authors to consider following recommendations carefully and resubmit their manuscript after through revision:
1. “The results showed that polymer concentration had the greatest influence on PAN nanofiber diameter.” – What is the novelty of this work? Similar conclusion was achieved by a lot of researchers on PAN or other known polymers using neural network model or other models before. Some examples are Journal of Materials Processing Technology 209(7):3156-3165 and Materials & Design, Volume 194, September 2020, 108902. Please explain in the text.
2. Also, compared to the references 17-20, what is unique in this study? Please discuss in the Introduction section.
3. What was the PAN molecular weight used in this study?
4. “PAN were prepared at concentrations …” – Please correct the sentence. Did you mean PAN solutions?
5. “The electrospinning voltage, distance between the spinneret and the collector, and flow rate of the spinning solution were optimized.” – Please refer to the supplementary file here for detailed parameter values.
6. There are 3 well known factors that could impact the diameter and morphology of the nanofibers. Those are the needle diameter, ambient factors such as % relative humidity and temperature. Why did not the authors consider those parameters? Please explain in the text.
7. Fig. 5: Please revise the Y-axis range (up to 0.15 should be sufficient).
8. Effect of PAN solution concentration on the fiber diameter is also closely influenced by the molecular weight of the PAN. Without addition of this variable in the model, this study seems incomplete. I would strongly recommend the addition of this variable for further consideration.
Round 2
Reviewer 1 Report
The authors have sufficiently addressed all points raised by the reviewer, and the infos & discussion added significantly improved the quality of the manuscript. I have no further restrictions to published the manuscript in the revised version.
Quality of English language is goos, only minor editing/grammar check is needed.
Reviewer 2 Report
Acceptance is recommended.
Please check for the affiliation 4 and 5. Is there a university name missing?